# Increased Prevalence of EBV Infection in Nasopharyngeal Carcinoma Patients: A Six-Year Cross-Sectional Study

**DOI:** 10.3390/cancers15030643

**Published:** 2023-01-19

**Authors:** Abdullah E. Al-Anazi, Bader S. Alanazi, Huda M. Alshanbari, Emad Masuadi, Maaweya E. Hamed, Iman Dandachi, Abdulrahman Alkathiri, Atif Hanif, Islam Nour, Hanadi Fatani, Hadel Alsaran, Fahad AlKhareeb, Ali Al Zahrani, Abdullah A. Alsharm, Saleh Eifan, Bandar Alosaimi

**Affiliations:** 1Comprehensive Cancer Center, King Fahad Medical City, Riyadh Second Health Cluster, Riyadh 11525, Saudi Arabia; 2Botany and Microbiology Department, College of Science, King Saud University, Riyadh 12372, Saudi Arabia; 3Research Center, King Fahad Medical City, Riyadh Second Health Cluster, Riyadh 11525, Saudi Arabia; 4Department of Mathematical Sciences, College of Science, Princess Nourah bint Abdulrahman University, P.O. Box 84428, Riyadh 11671, Saudi Arabia; 5Institute of Public Health, College of Medicine, UAE University, Al Ain 15551, United Arab Emirates; 6Pathology Clinical Laboratory Medicine Administration, King Fahad Medical City, Riyadh Second Health Cluster, Riyadh 11525, Saudi Arabia

**Keywords:** EBV, NPC, epidemiology, Saudi Arabia, prevalence, genotyping

## Abstract

**Simple Summary:**

Epstein Barr Virus (EBV) is associated with at least 1% of global cancers including nasopharyngeal carcinoma (NPC). Studies on the molecular epidemiology of EBV should improve the understanding of NPC prognosis. Retrospectively, we collected demographic and clinical data for 146 NPC patients over a 6-year period between 2015 and 2020. We found a high prevalence of 96% of EBV infection in NPC patients with a predominance of genotype I detected in 73% of NPC samples. Although NPC had metastasized to 16% of body sites, it was not associated with EBV infection, except for lungs. Three-quarters of NPC patients were in the advanced stages of cancer and the overall survival (OS) mean time was 5.59 years. We found an increased prevalence of EBV infection in NPC patients higher than previously thought with a predominance of EBV genotype I. A future multi-center study with a larger sample size is needed to assess the true burden of EBV-associated NPC.

**Abstract:**

Epstein Barr Virus (EBV) is implicated in the carcinogenesis of nasopharyngeal carcinoma (NPC) and currently associated with at least 1% of global cancers. The differential prognosis analysis of NPC in EBV genotypes remains to be elucidated. Medical, radiological, pathological, and laboratory reports of 146 NPC patients were collected retrospectively over a 6-year period between 2015 and 2020. From the pathology archives, DNA was extracted from tumor blocks and used for EBV nuclear antigen 3C (EBNA-3C) genotyping by nested polymerase chain reaction (PCR). We found a high prevalence of 96% of EBV infection in NPC patients with a predominance of genotype I detected in 73% of NPC samples. Histopathological examination showed that most of the NPC patients were in the advanced stages of cancer: stage III (38.4%) or stage IV-B (37.7%). Only keratinized squamous cell carcinoma was significantly higher in EBV negative NPC patients compared with those who were EBV positive (OR = 0.01, 95%CI = (0.004–0.32; *p* = 0.009)), whereas the majority of patients (91.8%) had undifferentiated, non-keratinizing squamous cell carcinoma, followed by differentiated, non-keratinizing squamous cell carcinoma (7.5%). Although NPC had metastasized to 16% of other body sites, it was not associated with EBV infection, except for lung metastasis. A statistically significant reverse association was observed between EBV infection and lung metastasis (OR = 0.07, 95%CI = (0.01–0.51; *p* = 0.008)). Although 13% of NPC patients died, the overall survival (OS) mean time was 5.59 years. Given the high prevalence of EBV-associated NPC in our population, Saudi could be considered as an area with a high incidence of EBV-associated NPC with a predominance of EBV genotype I. A future multi-center study with a larger sample size is needed to assess the true burden of EBV-associated NPC in Saudi Arabia.

## 1. Introduction

Nasopharyngeal carcinoma (NPC) is a malignancy that arises in the lining of the nasopharyngeal mucosa [1]. It is a distinct type of head and neck cancer (HNC) that has a complicated and poorly known pathogenesis with several predisposing factors [2]. In 2018, there were 129,079 new NPC cases worldwide, which led to 72,987 fatalities; approximately 70% of these new cases were from Eastern and South Eastern Asia [3]. In Saudi Arabia, HNC accounts for 6% of all malignancies diagnosed annually. Among these, NPC constitutes 33%, with an annual age-standardized incidence of 0.25 per 10,000 in males and 0.08 per 10,000 in females [4]. Among Arab countries, Saudi Arabia has the second highest prevalence of nasopharyngeal carcinoma [5].

The World Health Organization (WHO) divides NPC into three types: non-keratinizing, keratinizing, and basaloid squamous cell carcinoma (SCC). Non-keratinizing tumors are further classified as undifferentiated or differentiated [6]. Environmental, pathogen-mediated, and genetic factors, as well as lifestyle (smoking and eating nitrosamine-containing canned goods), can increase the likelihood of developing NPC [2]. Moreover, infection with EBV has been reported as a major risk factor for this type of malignancy [7].

It was demonstrated that high levels of circulating EBV DNA were associated with poorer response to treatment and higher rates of distant metastasis and mortality in NPC patients [8]. EBV is a gammaherpesvirus 4 that is commonly found in the human population where it is transmitted orally via saliva containing infected epithelial cells [9]. It has been estimated that more than 95% of individuals become infected during their childhood and early adolescence [10]. Primary infection with EBV is often asymptomatic but it can, in some cases, cause mononucleosis, a febrile illness characterized by elevated viral loads and exaggerated virus-induced immune responses [11]. When EBV progresses to latent infection, it can be associated with the development of cancer.

EBV is classified in group 1 of human carcinogens [12]. It was the first human oncogenic virus to be discovered and currently is the only human pathogen that can immortalize and transform cells in vitro [12]. EBV is a double-stranded DNA virus with an icosahedral capsid and a glycoprotein-containing envelope. It has a relatively large DNA genome (170–175 kbp) that comprises ˃94 ORFs that give rise to ˃100 proteins [13]. This virus has two genotypes, I and II, which have been differentiated based on the sequence heterogeneity of EBNA-2 and EBNA-3 [9,14]. Within each genotype, several strains are distinguished based on their genetic variation and high sequence divergence between both genotypes [15,16]. In Saudi Arabia, little is known about EBV detection, diagnosis, molecular analysis, or its association with cancer [17,18]. In one study of 25 Saudi patients with NPC, it was found that about 92% of the tumor specimens were EBV positive [19]. In another study of Hodgkin’s lymphoma cases, EBV was detected in 42 (28.6%) and 9 (30%) of Saudi and European cases, respectively [20]. AlDhahri et al., however, found that over a 7-year period, EBV was detected in 98.4% of samples derived from malignant cells whereas only 6.6% had latent EBV infection in the normal cells of NPC patients [21]. The association of EBV with clinical outcomes in NPC patients and the circulating genotypes was not previously explored in Saudi Arabia. The aim of this study was therefore to examine the association between EBV and demographic factors, as well as the clinical characteristics of NPC patients, in a major tertiary care center in Saudi Arabia over a six-year period. This was in addition to determining the genotype of EBV in this specific group of patients.

## 2. Materials and Methods

### 2.1. Ethics Statement

Ethical approval for collecting NPC patients’ data and FFPE samples was obtained from the Institutional Review Board (IRB) of King Fahad Medical City (KFMC), Riyadh, Saudi Arabia (20–785). Due to the retrospective nature of the study and the use of only anonymized leftover tissue without any personally identifiable data, signed informed consent was not required and was waived by the IRB. 

### 2.2. Study Design

The annual records of nasopharyngeal carcinoma (NPC) patients for 6 years from 2015–2020 were retrieved from the tumor registry at the Comprehensive Cancer Centre (CCC), King Fahad Medical City (370 cases). Patients were excluded if their tumor biopsy was taken from lymph nodes and not from the nasopharyngeal site (15 patients) and if their histopathological report does not contain an immunohistochemistry test for the presence or absence of EBV (209 patients). This resulted in 146 NPC patients that were included in the final analysis.

### 2.3. Patients’ Data Collection

KFMC is one of the largest medical complexes in the Middle East with a total capacity of 1200 beds. The demographic, as well as the clinical characteristics of 146 NPC patients from 2015 to 2020, were retrieved from their medical, radiological, pathological, and laboratory reports. Demographic data included age, gender, and nationality. Around three-quarters of these were males and the majority were of Saudi Arabian nationality. Clinical characteristics included: the presence of other comorbidities (such as diabetes mellitus and hypertension), tumor morphology, stage of cancer (I, II, III, IV-A, and IV-B), metastasis, site of metastasis, treatment protocol, treatment response, cancer recurrence, date of cancer diagnosis, date of last contact with the patient, and EBV status. All of the basic characteristics of the enrolled patients are shown in Table 1. Detailed characteristics of the enrolled patients are shown in Appendix A.

### 2.4. FFPE Collection and Histopathological Examination 

Among the 146 patients included in the final analysis, only 53 paraffin-embedded (FFPE) tumor blocks were available and collected from the pathology archives at KFMC. Thin slices were cut from the FFPE blocks, placed on a glass slide, and stained with hematoxylin and eosin (H&E). They were examined under a microscope to determine the tumor morphology of NPC patients. Morphologies comprised keratinizing squamous cell carcinoma and non-keratinizing squamous cell carcinoma. The latter included differentiated and un-differentiated types [6]. The EBV status of 146 NPC patients, i.e., positive or negative was obtained from the pathological reports provided by the diagnostic laboratory at KFMC.

### 2.5. DNA Extraction and Identification of EBV Genotype

The DNA of EBV was extracted from the 53 FFPE tissue blocks using the QIAamp^®^ DNA FFPE Tissue kit following the manufacturer’s instructions (QIAGEN, Hilden, Germany, cat # 56404). Extracted DNA was eluted in a 50 μL final volume and was then quantified using a NanoDrop spectrophotometer (NanoDrop™ 2000/2000c Spectrophotometers). The genotype of EBV in collected samples was determined as follows: extracted DNA was subjected to nested polymerase chain reaction (PCR) for the detection of EBV nuclear antigen 3C (EBNA-3C) gene. For the first PCR cycle, 12.5 µL of 1× DreamTaq Hot Start Green PCR master mix (Thermo Fisher Scientific, Carlsbad, CA, USA) was mixed with 1.5 µL (10 µM) forward primer, 1.5 µL (10 µM) reverse primer, 4.5 µL nuclease-free water, and 5 µL of DNA template in a sterile Eppendorf tube. The following primers set was used: EBNA-3C-F-5′-GAGAAGGGGAGCGTGTGTTGT-3′ and EBNA-3C-R-5′-GGCTCGTTTTTGACGTCGGC-3′ [22]. Expected product sizes were 153 bp and 246 bp for genotypes 1 and 2, respectively. PCR thermal cycling conditions were as follows: initial denaturation at 95 °C for 3 min, followed by 30 cycles of 95 °C for 30 s, annealing at 60 °C for 30 s, 72 °C for 1 min, and 72 °C for 15 min as a final extension. The second PCR run was carried out using EBNA-3C-F’-5′-TCATAGAGGTGATTGATGTT-3′ and EBNA-3C-R’-5′-ATGTTTCCGATGTGGCTTAT-3′ [22] following the same protocol and conditions but with an annealing temperature of 50 °C. The expected product sizes for this round were 75 bp for genotype 1 and 168 bp for genotype 2. We included an EBV-positive sample which was confirmed by PCR and RT-PCR in the molecular virology laboratory at KFMC, as well as the housekeeping gene “GAPDH” as an internal control (GAPDH-F-GTATTGGGCGCCTGGTCACC and GAPDH-R-CGCTCCTGGAAGATGGTGATGG). PCR products were run on a 2% agarose gel in Tris-acetate-EDTA (TAE) Buffer 1× and visualized under UV light using the BioRad ChemiDoc Touch Imaging System (Bio-Rad, Hercules, CA, USA).

### 2.6. Statistical Analysis

Data were entered and analyzed using the statistical package IBM SPSS Statistics for Windows (Version 26.0, IBM Corp, Armonk, NY, USA). Categorical data were presented as frequencies and percentages. The odds ratio (OR) with a 95% confidence interval was used as a measure of association. The Wald statistic with the Chi-square test was used to assess the association between the EBV status and other categorical variables. The Kaplan–Meier curve with a log rank (Mantel–Cox) test was used to compare the survival distribution between positive and negative EBV. All tests were considered significant if the *p*-value < 0.05.

## 3. Results

### 3.1. NPC Characteristics of the Studied Population

A total of 146 NPC patients were included. Most of the patients were either in stage III (38.4%) or in stage IV-B (37.7%) of cancer. In addition to the primary tumor, 23 (16%) of the NPC patients had at least one site of metastasis. Among these sites, bone (8.2%), followed by liver (5.5%), lungs (4.8%), and other body sites (6.2%) were the most common. The majority of patients (91.8%) had undifferentiated, non-keratinizing squamous cell carcinoma, followed by differentiated, non-keratinizing squamous cell carcinoma and keratinized squamous cell carcinoma (Figure 1, Table 2). Only 6.2% of the patients had recurrent cancer. In terms of treatment, 24% were under concurrent chemoradiation therapy (CCRT) and 60.3% under induction/CCRT. The majority of the NPC patients (84.2%) had a good response to the treatment. Only 13% died, with 63.2% of deaths due to nasopharyngeal carcinoma. EBV was positive in 140 (96%) NPC patients. The characteristics and treatment of the studied cohort are summarized in Table 2.

### 3.2. EBV Infection and NPC Patients’ Baseline Characteristics

Our data show that there was no association between EBV-positive infection and NPC patients’ demographic and clinical data such as age, disease stage, treatment protocol, and presence of comorbidities. However, the incidence of EBV was significantly higher in NPC male patients compared with females (OR = 16.21, 95%CI = (1.83–143; *p* = 0.012)) (Table 3).

### 3.3. EBV Infection and NPC Patients’ Clinical Characteristics

The association of EBV status and patients’ clinical characteristics showed no statistical significance. EBV was positivity associated with better treatment response as well as advanced disease stage, although this association was not statistically significant for both observations. Similarly, metastasis of nasopharyngeal carcinoma to other body sites including liver and bone was not associated with EBV positivity, except for lung metastasis. NPC patients with EBV had a significantly lower rate of lung metastasis compared with those without EBV (OR = 0.07, 95%CI = (0.01–0.51; *p* = 0.008)) (Table 4). Furthermore, our data showed that undifferentiated non-keratinizing squamous cell carcinoma was higher in positive EBV patients (97%) compared with EBV-negative patients (3%) but this was without statistical significance. There was no basaloid squamous cell carcinoma. However, keratinized squamous cell carcinoma was significantly higher in EBV-negative NPC patients compared with those who were EBV positive (OR = 0.01, 95%CI = (0.004–0.32; *p* = 0.009)) (Table 4).

### 3.4. EBV Infection and NPC Patients’ DFS and Overall Survival

The mean time measured in months of disease-free survival (DFS) for all NPC patients was 5.80 years. NPC patients with negative EBV status tended to have a lower mean DFS time, but without statistical significance (mean = 3.88 years, 95%CI = (1.53–6.24) vs 5.85 years, 95%CI = (5.47–6.22); *p* = 0.166) (Figure 2A). On the other hand, the overall survival (OS) mean time of all NPC patients was 5.59 years. We observed that NPC patients with negative EBV status also tended to have a lower mean survival time but this was not statistically significant (mean = 3.77 years, 95%CI = (1.82–5.72) vs. 5.66 years, 95%CI = (5.2–6.12); *p* = 0.132) (Figure 2B).

### 3.5. EBV Genotypic Analysis

All the 53 collected FFPE blocks of NPC patients were subjected to genotyping. (51 were from EBV-positive NPC patients and 2 were from EBV-negative NPC patients). Total DNA mean concentrations and purities were 264 ng/µL and 2, respectively. EBV genotyping using the EBNA-3C gene showed the amplification of 153 bp and 246 bp fragments in the first round of PCR; the latter are characteristics of genotype I and II, respectively (Figure 3A). In the second round, amplicons of 75 bp and 168 bp in size were obtained, displaying type I and type II EBV genotypes, respectively (Figure 3B). EBV genotype I was detected in 37 (73%) and genotype II was detected in 1 (2%) of the NPC samples, whereas 13 samples (25%) could not be genotyped because PCR reaction yielded insufficient amplicon.

## 4. Discussion

The incidence of head and neck malignancies increased globally by 45% between 2009 and 2019 from 121,650 to 176,500 cases [23]. Nasopharyngeal carcinoma, a common type of HNC, is an epithelial tumor that arises from the Rosenmuller pit in the nasopharynx [24]. EBV is globally known to be directly implicated in carcinogenesis [25]. This virus is currently associated with 1% of global cancers, including nasopharyngeal carcinoma [25,26]. In accordance with the global EBV infection rates that affect more than 95% of individuals worldwide [10], our study found that EBV infection was observed in 96% of NPC patients. Its association with more than 95% of NPC cases was also reported in areas with a high incidence of NPC, such as Eastern and Southern Asian countries and some areas in the Middle East [27]. In low-incidence areas of NPC, EBV association is as low as 75% [27]. It was estimated that around 84.6% of all NPC cases worldwide could be associated with EBV infection [28]. Interestingly, our study showed a higher prevalence of EBV infection in NPC patients than in other studies, including those in Finland (62%) [29], Sudan (61.3%) [30], and Japan (63%) [31], but is comparable with the prevalence in Turkey (87%) [32]. Therefore, Saudi Arabia could be considered an area with a high incidence of EBV-associated NPC.

EBV-associated NPC occurrence is increasing in Saudi Arabia due to population exposure to several risk factors. A family history of NPC, chronic respiratory tract conditions, exposure to air pollutants, a higher intake of preserved food, and tobacco smoking are the associated risk factors [5]. We have found that the male gender was significantly associated with positive EBV infection in NPC patients. This is not surprising since in our cohort, the male gender was predominant over the female one (108 vs. 38). Similar findings were reported by other studies suggesting that males have a higher positive EBV occurrence with NPC than females [30,33,34,35]. The predominance of males with positive EBV in NPC patients can in part be explained by differences in environmental factors such as smoking, as well as hazardous occupational exposures [36]. The true reason behind this higher incidence, however, remains to be elucidated.

Although NPC is known to have the highest metastatic potential [37], in our study, only 16% of NPC patients had cancer metastasis. The differential prognosis analysis of nasopharyngeal carcinoma in EBV-infected versus non-infected NPC patients showed a reverse association between EBV infection and lung metastasis which meant lower rates of lung metastasis in NPC-EBV-positive subjects. Several studies have shown that NPC distant metastasis is one of the main causes of patients’ reduced survival rates with bone, liver, and lungs being the most common metastatic body sites reported in NPC patients [38]. The absence of a significant difference in metastasis between EBV-positive and negative NPC patients could in part explain the absence of a significant difference in the corresponding survival times between the two populations. Furthermore, the absence of a significant difference between infected and non-infected NPC patients could be attributed to the low number of EBV-negative subjects that rendered the difference in clinical outcomes statistically insignificant. Selective lung metastasis in our study could be explained by a complex multistep process known as “metastatic organotropism” [39], which is regulated by the cross-talk and interactions between intrinsic properties of cancer cells and the host organ microenvironment [40]. Further mechanistic studies on the distribution of distant metastases of EBV-associated NPCs to certain organs will improve our understanding of determining the regulatory mediators of organ-specific metastasis.

Based on differences in genetic variations, EBV was classified into genotype I and genotype II [15]. Our study revealed the predominance of EBV genotype I (97%, 37/38) in NPC patients, with only one being genotype II. This finding is not surprising, as genotype I is the most common type reported globally with higher frequencies being observed in Asia, Europe, and the Americas [27]. In contrast, genotype II is mostly detected in Africa and New Guinea [27]. In a study conducted in Ghana, EBV genotype II was found to be dominating in nasopharyngeal biopsies [41]. In studies from European and Asian countries such as Serbia and China, however, genotype I was the most common in cancer patients [27,42]. Globally, neither type I or type II has been linked to a specific disease. For example, in Banko et. al.’s study from China, EBV type I dominated in patients with leukemia as well as those with myelodysplastic syndrome [42]. Likewise, in a study conducted in healthy blood donors from different nationalities in Qatar, EBV genotype I also predominated [43]. It is worth noting in our study that thirteen samples could not be genotyped because PCR reaction yielded insufficient amplicon. One possible explanation could be the degradation or fragmentation of the viral DNA inside the block, possibly due to a stringent paraffin fixation procedure or long-term storage. Prolonged formalin fixation causes proteins as well as nucleic acid crosslinking. This is in addition to random breakages in the nucleotide sequences [44]. 

Our study has two limitations. The first we encountered is the low number of EBV-negative NPC samples found during the six-year period at the cancer registry at KFMC. This low number could have affected the statistical significance of several variables and clinical outcomes explored in our study. The second limitation was the unavailability of the majority of the FFPE tissue blocks. This could be due to either tumor exhaustion in the block for diagnostic purposes or that the patient had taken the block to another hospital.

## 5. Conclusions and Future Perspectives

Although this study reveals a higher prevalence of EBV infection in NPC patients, a multi-center study with larger sample size is needed to assess the true burden of this virus in this type of cancer and whether Saudi could be considered as an area with a high incidence of EBV-associated NPC. Our study showed no statistically significant difference in the prognosis between EBV-infected and non-infected NPC patients, except for lower rates of lung metastasis in NPC-EBV-positive subjects. Further mechanistic studies on the distribution of distant metastases of EBV-associated NPCs to certain organs will improve our understanding on determining the regulatory mediators of organ-specific metastasis.

## Figures and Tables

**Figure 1 cancers-15-00643-f001:**
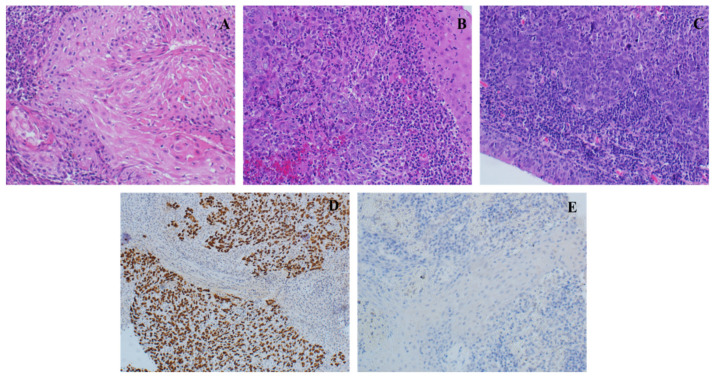
Histopathological examination of nasopharyngeal carcinoma observed under 20× magnification. (**A**) Keratinizing squamous cell carcinoma. (**B**) Non-keratinizing differentiated squamous cell carcinoma. (**C**) Non-keratinizing undifferentiated squamous cell carcinoma. (**D**) EBER in situ hybridization stain for EBV positive. (**E**) EBER in situ hybridization stain for EBV negative.

**Figure 2 cancers-15-00643-f002:**
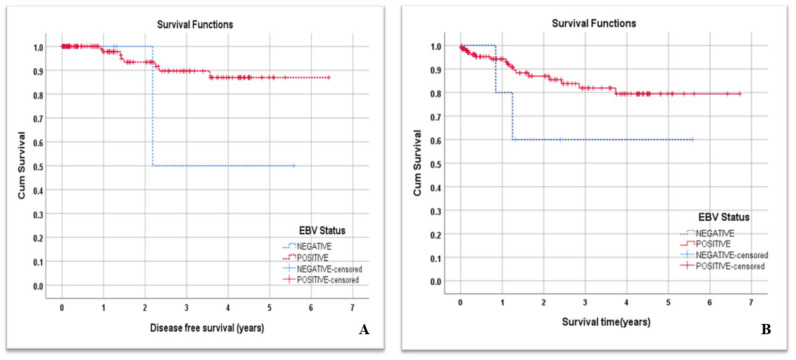
(**A**) Disease-free survival; (**B**) overall survival curves of all patients by EBV status.

**Figure 3 cancers-15-00643-f003:**
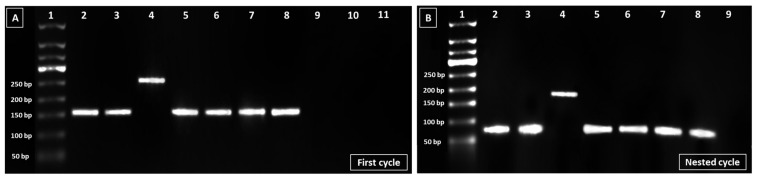
(**A**) First round PCR product of EBNA-3C gene. Lane 1 is a 500 bp DNA Ladder (50–500 bp). Lanes 2, 3, and 5–8 represent EBV type I amplicons “EBNA-3C, 153 bp”. Lane 4 represents EBV type II amplicons “EBNA-3C, 246 bp”. Lane 9 corresponds to the negative control (nuclease-free water with master mix). Lanes 10 and 11 correspond to the EBV-negative control samples. (**B**) Second round of the PCR product of the EBNA-3C gene. Lane 1 is a 500 bp DNA Ladder (50–500 bp). Lanes 2, 3, and 5–8 represent EBV type I amplicons “EBNA-3C, 75 bp”. Lane 4 represents EBV type II amplicons “EBNA-3C, 168 bp”. Lane 9 corresponds to the negative control (nuclease-free water with master mix).

**Table 1 cancers-15-00643-t001:** Patients’ demographic information and medical history.

Patients’ Characteristics	Variables	Number	Percentage
Age (years)	≤30	20	13.7%
	31–40	25	17.1%
	41–50	40	27.4%
	51–60	37	25.3%
	≥61	24	16.4%
Gender	Male	108	74.0%
	Female	38	26.0%
Nationality	Saudi Arabian	116	79.5%
	Non-Saudi Arabian	30	20.5%
Comorbidities	Hypertension	23	15.7%
	Diabetes mellitus	25	17%
	Other diseases	25	17%
EBV Status	Negative	6	4%
	Positive	140	96%
Patient’s Status	Alive	127	87.0%
	Deceased	19	13.0%
Cause of death	NPC	12	63.2%
	Not NPC	7	36.8%

**Table 2 cancers-15-00643-t002:** Patients’ cancer characteristics.

Patients’ Characteristics	Variables	Number	Percentage
Stage of cancer	I	2	1.4%
	II	10	6.8%
	III	56	38.4%
	IV-A	55	37.7%
	IV-B	23	15.8%
Morphology	Keratinized, squamous cell carcinoma	1	0.7%
	Differentiated, non-keratinizing Squamous cell carcinoma	11	7.5%
	Undifferentiated, non-keratinizing squamous cell carcinoma	134	91.8%
TNM staging (T)	T0	2	1.4%
	T1	35	24%
	T2	24	16.4%
	T3	37	25.3%
	T4	48	32.9%
TNM staging (N)	N0	12	8.2%
	N1	25	17.1%
	N2	68	46.6%
	N3	41	28.1%
TNM staging (M)	M0	123	84%
	M1	23	16%
Metastasis	Yes	23	15.8%
	No	123	84.2%
Recurrent cancer	No	137	93.8%
	Yes	9	6.2%
Treatment protocol	No treatment	9	6.2%
	RT	2	1.4%
	CCRT	35	24.0%
	Induction + CCRT	88	60.3%
	Palliative	12	8.2%
Treatment Response	No treatment	9	6.2%
	Good response	123	84.2%
	Poor response	14	9.6%
Cause of 19 deaths	NPC	12	63.2%
	Not NPC	7	36.8%

RT = radiation therapy, CCRT = concurrent chemoradiation therapy, T = tumor (size), N = node (nearby lymph nodes that have cancer), and M = metastasis (cancer has metastasized from the primary site).

**Table 3 cancers-15-00643-t003:** Association between EBV status and patients’ baseline characteristics.

Variable	Categories	EBV Status							
		Negative		Positive		OR	95%CI for OR		
		*N*	%	*N*	%		Lower	Upper	*p*-Value
Age (years)	≤30	1	5.0	19	95.0	1.73	0.15	20.58	0.665
	31–40	1	4.0	24	96.0	2.18	0.19	25.77	0.536
	41–50	1	2.5	39	97.5	3.55	0.30	41.36	0.313
	51–60	1	2.7	36	97.3	3.27	0.28	38.24	0.345
	≥61	2	8.3	22	91.7	1.00			
Gender	Male	1	0.9	107	99.1	16.21	1.83	143.74	0.012 *
	Female	5	13.2	33	86.8	1.00			
Nationality	Saudi	6	5.2	110	94.8	0.28	0.02	5.07	0.389
	Non-Saudi	0	0.0	30	100.0	1.00			
DM	No	6	5.0	115	95.0	1			0.48
	Yes	0	0.0	25	100.0	2.78	0.16	52.60	
HTN	No	5	4.1	118	95.9	1			0.95
	Yes	1	4.3	22	95.7	0.93	0.10	8.37	
Other diseases	No	5	4.1	116	95.9	1			0.98
	Yes	1	4.0	24	96.0	1.03	0.12	9.25	
Treatment protocol	RT	0	0.0	2	100.0	0.88	0.03	29.14	0.944
	CCRT	2	5.7	33	94.3	2.06	0.17	25.68	0.574
	Induction + CCRT	3	3.4	85	96.6	3.54	0.33	38.13	0.297
	Palliative	0	0.0	12	100.0	4.41	0.16	121.70	0.381
	No treatment	1	11.1	8	88.9	1.00			

*N* = Number, DM = diabetes mellitus, HTN = hypertension, Mets = metastasis, RT = radiation therapy, CCRT = concurrent chemoradiation therapy, OR = odd ratio, and CI = confidence interval. * Indicate significant *p*-values (*p* < 0.05).

**Table 4 cancers-15-00643-t004:** Association between EBV status and patients’ clinical characteristics.

Variable	Categories	EBV Status							
		Negative		Positive		OR	95%CI for OR		
		*N*	%	*N*	%		Lower	Upper	*p*-Value
Stage group	I	1	50.0	1	50.0	0.10	0.01	2.18	0.141
	II	0	0.0	10	100.0	2.44	0.11	55.56	0.576
	III	2	3.6	54	96.4	2.57	0.34	19.46	0.360
	IV-A	1	1.8	54	98.2	5.14	0.44	59.77	0.191
	IV-B	2	8.7	21	91.3	1.00			
Bone metastasis	No	5	3.7	129	96.3	1			0.45
	Yes	1	8.3	11	91.7	0.43	0.05	3.97	
Liver metastasis	No	5	3.6	133	96.4	1			0.25
	Yes	1	12.5	7	87.5	0.263	0.03	2.57	
Lung metastasis	No	4	2.9	135	97.1	1			0.008 *
	Yes	2	28.6	5	71.4	0.07	0.01	0.51	
Other sites of Metastasis	No	6	4.4	131	95.6	1			0.97
	Yes	0	0.0	9	100.0	0.94	0.05	17.95	
Morphology	Keratinized, squamous cell carcinoma	1	100.0	0	0.0	0.01	0.00	0.32	0.009 *
	Differentiated, non-keratinizing squamous cell carcinoma	1	9.1	10	90.9	0.308	0.031	3.02	0.312
	Undifferentiated, non-keratinizing squamous cell carcinoma	4	3.0	130	97.0	1.00	-	-	-
Treatment Response	Good response	3	2.4	120	97.6	5.00	0.47	53.70	0.184
	Poor response	2	14.3	12	85.7	0.75	0.06	9.72	0.826
	No treatment	1	11.1	8	88.9	1.00	-	-	-

*N* = Number, OR = odd ratio, CI = confidence interval. * Indicate significant *p*-values (*p* < 0.05).

## Data Availability

The data sets generated during and/or analyzed during the current study are available from the corresponding author upon reasonable request.

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
