# Peer review of "Increased Prevalence of EBV Infection in Nasopharyngeal Carcinoma Patients: A Six-Year Cross-Sectional Study"

_cancers, 2023, doi:10.3390/cancers15030643_

Round 1
Reviewer 1 Report
The paper entitled ‘Increased prevalence of EBV infection in Nasopharyngeal Carcinoma Patients: molecular epidemiology over a six-year period’ investigates the impact of Epstein Barr Virus on the clinical characteristics of NPC patients. The topic is interesting, and for sure is important, as prevalence of EBV is high.
Presented paper is neither pathbreaking nor innovative, as similar papers had been already published for example:
· Shahani, T., Makvandi, M., Samarbafzadeh, A., Teimoori, A., Ranjbar, N., Nikakhlagh, S., ... & Haghi, A. (2017). Frequency of Epstein Barr virus type 1 among nasopharyngeal carcinomas in Iranian patients. Asian Pacific Journal of Cancer Prevention: APJCP, 18(2), 327.
· Argirion, I., Zarins, K. R., Ruterbusch, J. J., Vatanasapt, P., Sriplung, H., Seymour, E. K., & Rozek, L. S. (2020). Increasing incidence of Epstein‐Barr virus–related nasopharyngeal carcinoma in the United States. Cancer, 126(1), 121-130.
· Xiong, G., Zhang, B., Huang, M. Y., Zhou, H., Chen, L. Z., Feng, Q. S., ... & Zeng, Y. X. (2014). Epstein-Barr virus (EBV) infection in Chinese children: a retrospective study of age-specific prevalence. PLoS One, 9(6), e99857.
· Ruuskanen, M., Irjala, H., Minn, H., Vahlberg, T., Randen‐Brady, R., Hagström, J., ... & Leivo, I. (2019). Epstein‐Barr virus and human papillomaviruses as favorable prognostic factors in nasopharyngeal carcinoma: a nationwide study in Finland. Head & Neck, 41(2), 349-357.
Nevertheless, it may be have some value for the clinician. The study concerns the population with relatively high incidence of EBV and NPC, therefore it may be particularly interesting. However, in my opinion it cannot be published in the present form, as there are too may unclear issues and it demands explanation.
Title – the title is slightly deceptive. You have mentioned about molecular epidemiology, it is definitely not the core of the paper (about 38 analyzed samples). You have just prepared PCR without sequencing and further analysis (for example phylogenetic relations between variants). Edit the title to better display the content of the paper.
Material and methods
Line 104 – in this line you should include the number of patients. I know that it has already been explained in abstract, but it is also mandatory in material and methods section.
Table 1 - Provide supplementary materials giving more details about characteristics, what exactly means non-Saudi ? What was the cause of death not NPC death ? What means other diseases (n=25 it is relatively numerous group, so it will be interesting to give additional information)?
Line 112 – How many samples were analyzed with EBER test ?
Line 117 – Could you provide more details about the EBER test ?
Line 121 – How many samples were subjected to DNA extraction ? Each sample which was confirmed as positive by EBER test ?
Line 124 – Provide information about DNA concentrations and purity in the results section.
Line 126 – Did you include any internal control such as B-globin to ascertain the presence of human genomic DNA
Line 132 -What was the expected size of PCR products? Mention about positive and negative control. In the results, I see only negative control.
Results
Lines 152-154 - These values do not sum up to 16%
Line 157 – I think that you should also include the reference to table 2, as you write about frequency, which is actually presented in the table.
Line 161 – Cancer
Table 2 – explain abbreviation TNM
Line 209-216 – Title includes molecular epidemiology, and you have just prepared the nested PCR without further sequencing. Of course, you have detected the genotypes, but in my opinion it is not enough to analyze molecular epidemiology, especially without proper controls (positive control, internal control)
Line 210 – Data from the tables indicate that 140 patients were EBV positive so why in the line 210, there is n=51 ?
Line 216 – again without proper controls, it is very difficult to assess if there was too low viral load or reaction was not properly optimized. It is difficult to prepare the reliable molecular epidemiology without control and with 25% samples consider as poor quality amplicons. If we sum up all samples, you have prepared the molecular epidemiology on the basis of 38 positive samples, whereas you have 140 positive . Explain, why ?
Discussion
Actually, this section is quite interesting. Nevertheless, I have some doubts, about lines 281-287. Degradation of viral DNA may be one explanation, but I encourage Authors to discuss this issue more extensively. You have amplified relatively short sequences, therefore despite degradation we may expect some PCR products. Is it possible that viral load was too low ? May there occur the polymorphisms in the primer binding sites ?
Here we see again how positive and internal control is important. Amplifying human DNA (for example B-globin) you can confirm or rule out the worse quality of amplicons. Is it possible to repeat analysis with primers targeting some human DNA ?
Lines 288-293 I am really appreciate that Authors are aware of the limitations of their study, but this limitations significantly decrease the merit of the paper. We get information about the lower number of samples at the last sentence of discussion. These information should be included in the material and methods section. Please explain how many samples were analyzed in each step. As I understand, you have clinical data for whole group – 146, but how many of them were subjected to EBER test ? How many were analyzed in PCR test? 38 out of 146 ?
Decision
At first the paper seemed to be quite interesting but there are too many question marks. The part of the title – molecular epidemiology over a six year period is not justified. Epidemiology is based on the 38 samples out of 146, there were neither positive nor internal control and this is serious drawback in designing the study. I see some potential in this study, but a lot of things should be explained (number of samples on each stage of testing, lack of controls). I decided that Authors should get the chance to address the comments, therefore I recommend the major revision.
Reviewer 2 Report
The manuscript entitled „Increased prevalence of EBV infection in Nasopharyngeal Carcinoma Patients: molecular epidemiology over a six-year period“ (cancers-2132666) by Alanazi et al. investigates the frequency of EBV infection in 146 human NPC FFPE specimen and the respective correlations to clinical parameters. It is well written and the results contribute to a better understanding of EBV and its role in NPCs.
However, before this article is acceptable for publication several major and minor comments must be addressed.
Major comments:
As already mentioned by the authors the EBV negative group (n=6) is way too small for reliable statistical analyses.
Therefore, the analyses of the disease-free survival and overall survival could be more interesting by analyzing the data for EBV-positive males and EBV-positive females separately and also for other subgroups leading to the next critical point.
There is a strong gender imbalance per se in the analyzed cohort of 146 patients with 108 males and 38 females. It must be addressed by the authors also in regard to the only statistically significant difference in EBV frequency of male and female NPCs…
Also mentioned by the authors at the end of the manuscript: unavailability of the majority of FFPE tissue blocks.
This must also be mentioned at the results part when surprisingly only 51 specimen are analyzed by nested PCR.
Minor comments:
1. Nested PCR: An EBV-negative control is missing – if lane #9 is water control (see Figure 3).
2. Check spelling „tumour“ and „tumor“
3. Page 4 line 172: „…The cencer characteristics…“
Round 2
Reviewer 1 Report
Thank you for the revised version of the paper. I have to admit that the Authors addressed most of my comments and many doubts have been explained. I think that we all have to agree that paper has some limitations concerning the small number of tested samples and the gender imbalance. Nevertheless, I see the potential of the paper and its clinical value. Therefore, I think that the paper meets requirements of publication after minor revision.
The present title is more suitable and reflects the matter of the paper.
The section – materials and methods now is more clear. The good decision was to include the study design subsection, as it gives better insight into the whole analysis. I also appreciate the supplementary materials provided by the Authors.
Few things. At first, in line 233, you wrote – “EBV DNA mean concentrations and purities were 264 ng/μl and 2, respectively” . I think that it is not precise statement. As I think, you measured total DNA concentration (host DNA and EBV DNA), not only viral DNA.
The second issue is matter of controls. It is good to know that both internal and positive control were included but the information – ‘We included an EBV positive 238 sample which was confirmed by PCR and RT-PCR in the molecular virology laboratory 239 at KFMC, as well as housekeeping gene “GAPDH” as an internal control.’ should be included in materials and method section, not in the results. GAPDH is quite understandable choice. After you move this part to material and methods, please include the sequence of primers targeting gene fragment and the thermal profile of reaction to make study perfectly repeatable.
Correct the marker size and sample numbers in part A of Figure 3 to make it more visible.
I think that paper may be published after this minor revision.
Hope I was helpful.
Author Response
Dear Reviewer,
I appreciate the information and valuable comments you have shared, I also value the insights and guidance you provided. Thanks for your vote of confidence and kind complements on the paper, I am very grateful for your consideration. You comments have been addressed and the corrections have been applied to the manuscript.
Reviewer 2 Report
All comments were addressed by the authors. The manuscript is now acceptable for publication.
Author Response
Dear Reviewer,
I appreciate the information and valuable comments you have shared, I also value the insights and guidance you provided. Thanks for your vote of confidence and kind complements on the paper, I am very grateful for your consideration.
Round 3
Reviewer 1 Report
I think that paper meets requirements of publication. All suggested corrections have been made.